# Achieving Health Security and Threat Reduction through Sharing Sequence Data

**DOI:** 10.3390/tropicalmed4020078

**Published:** 2019-05-14

**Authors:** Kenneth Yeh, Jeanne Fair, Helen Cui, Carl Newman, Gavin Braunstein, Gvantsa Chanturia, Sapana Vora, Kendra Chittenden, Ashley Tseng, Corina Monagin, Jacqueline Fletcher

**Affiliations:** 1MRIGlobal, Gaithersburg, MD 20878, USA; 2Los Alamos National Laboratory, Los Alamos, NM 87545, USA; jmfair@lanl.gov (J.F.); hhcui@lanl.gov (H.C.); 3Defense Threat Reduction Agency, Fort Belvoir, VA 22060, USA; carl.i.newman.civ@mail.mil (C.N.); gavin.m.braunstein.civ@mail.mil (G.B.); 4National Center for Disease Control and Public Health, Tbilisi 0198, Georgia; gvantsa.chanturia@ncdc.ge; 5Department of State, Washington, DC 20520, USA; VoraSR@state.gov; 6USAID, Bureau of Global Health, Arlington, VA 22202, USA; kchittenden@usaid.gov; 7Department of Epidemiology, Columbia University, Mailman School of Public Health, New York, NY 10032, USA; at3346@cumc.columbia.edu; 8One Health Institute, University of California, Davis, CA 95616, USA; cmonagin@gmail.com; 9National Institute for Microbial Forensics & Food and Agricultural Biosecurity, Oklahoma State University, Stillwater, OK 74078, USA; jacqueline.fletcher@okstate.edu

**Keywords:** biosecurity, global health security, sequencing, scientific engagement, threat reduction

## Abstract

With the rapid development and broad applications of next-generation sequencing platforms and bioinformatic analytical tools, genomics has become a popular area for biosurveillance and international scientific collaboration. Governments from countries including the United States (US), Canada, Germany, and the United Kingdom have leveraged these advancements to support international cooperative programs that aim to reduce biological threats and build scientific capacity worldwide. A recent conference panel addressed the impacts of the enhancement of genomic sequencing capabilities through three major US bioengagement programs on international scientific engagement and biosecurity risk reduction. The panel contrasted the risks and benefits of supporting the enhancement of genomic sequencing capabilities through international scientific engagement to achieve biological threat reduction and global health security. The lower costs and new bioinformatic tools available have led to the greater application of sequencing to biosurveillance. Strengthening sequencing capabilities globally for the diagnosis and detection of infectious diseases through mutual collaborations has a high return on investment for increasing global health security. International collaborations based on genomics and shared sequence data can build and leverage scientific networks and improve the timeliness and accuracy of disease surveillance reporting needed to identify and mitigate infectious disease outbreaks and comply with international norms. Further efforts to promote scientific transparency within international collaboration will improve trust, reduce threats, and promote global health security.

“*Over the years, I have described international cooperation in addressing threats posed by weapons of mass destruction as a ‘window of opportunity.’ We never know how long that window will remain open. We must eliminate those conditions that restrict us or delay our ability to act*.”- Richard Lugar (1932–2019), former United States Senator and co-author of the Nunn–Lugar Cooperative Threat Reduction Program.

## 1. Introduction

While many governments fund international cooperation programs and use a range of technical approaches, scientific engagement and collaborations are arguably the most enduring. Effective international scientific engagement requires able and willing partners who bring complementary experience, knowledge, and problem-solving skills. Moreover, research and cooperative engagements performed with international partners reinforce weapons of mass destruction (WMDs) nonproliferation instruments including the Biological Weapons Convention (BWC) and the United Nations Security Council Resolution (UNSCR) 1540, which prohibit the use and spread of biological WMDs and are legally binding for all countries. This work also supports international frameworks for strengthening human and veterinary health systems such as the International Health Regulations (IHR) 2005 and the Global Health Security Agenda (GHSA).

The United States (US) Department of State (DOS) lists six US assistance programs that reduce biological threats around the world under the BWC’s Article X [1]. The three largest assistance programs are the DOS Biosecurity Engagement Program (BEP), the Defense Threat Reduction Agency (DTRA) Biological Threat Reduction Program (BTRP), and the US Agency for International Development (USAID) Global Health Security Agenda’s activities, including the Emerging Pandemic Threats Program (EPT). These programs provide assistance to over 38 partner nations worldwide, including the Former Soviet Union (FSU), Africa, Asia, the Middle East, and Latin America (Figure 1). Some of these engagements started after the dissolution of the Soviet Union with the goal of eliminating their weapons and redirecting their biological warfare programs. The various needs for assessing and securing biological WMD risks, addressing dual-use concerns, and surveilling infectious disease outbreaks are reflected in certain country and regional engagements (e.g., the FSU, Iraq, Africa, and Southeast Asia, respectively) [2,3]. Such engagements often reflect the partner country’s commitment to and interest in implementing objectives for those international nonproliferation instruments and related frameworks. One of the overarching mission objectives of these cooperative programs is to reduce the threat of infectious diseases whether the cause is accidental, intentional, or natural (Figure 2).

## 2. Methods: Design and Structure of Panel Discussion

Examples of interactive discussions such as panels, roundtables, and workshops that take place at scientific conferences are most effective when they juxtapose peer leaders with audiences to share new findings, discuss, and publish lessons learned together [4]. At the Sequencing, Finishing and Analysis for the Future (SFAF) conference held from 22 to 24 May 2018 in Santa Fe, New Mexico, USA, the audience learned of the work of the three major US assistance programs that focus on countering biological threats. An introductory presentation that addressed biosecurity and associated technological risks stimulated knowledge-sharing and utility. The panel addressed questions designed to highlight four areas: a) past successes from working cooperatively in global health security, b) current and future biosecurity concerns, c) challenges that remain for working together to combat infectious disease outbreaks globally, and d) how genomic sequencing can help to deter infectious disease outbreaks and be used to address biosecurity concerns. 

Jeanne Fair (Biosecurity and Public Health, Los Alamos National Laboratory) moderated the discussion with panelists Sapana Vora (BEP Team Lead), Gavin Braunstein (BTRP Science), Kendra Chittenden (USAID Emerging Threat Division Senior Infectious Disease Advisor), and Gvantsa Chanturia (BTRP partner-country scientist from the country of Georgia). The discussion topics were inspired by issues related to scientific transparency that emerged during the 2017 SFAF [5] such as the exchange of sequencing data in lieu of actual sample material. 

## 3. Discussion

US international cooperative engagement programs have been working with partner countries for nearly 25 years. While the initial geographic and political foci of the collaborations were primarily with the FSU, emerging diseases and the need to strengthen biosurveillance capabilities globally have led more recently to partnerships around the world. While panel members recognized that biosecurity concerns still remain, they emphasized that past successes keep the momentum of international partnerships moving forward in the face of continuing biosecurity threats that challenge efforts to keep support and collaborations in place.

The panel identified primary global biosecurity challenges related to the exchange of genomic information and sequencing data that include avoiding the misuse of pathogens, the proliferation and security of biobanks and repositories, the timely detection of emerging outbreaks, and infectious disease surveillance across participating countries worldwide. How can countries receiving cooperative engagement funding support themselves and their peers in meeting their IHR (2005) obligations while strengthening their biosecurity capabilities and capacity? Even after the past 25 years of US and international cooperation, significant challenges remain for safeguarding the access to and use of infectious pathogens, while new challenges such as discovering newly emerging pathogens and responding to disease outbreaks have emerged. Dual-use applications and novel technologies may lead to an increased risk of misuse or unintended consequences. With the increase of human, animal, and plant movements across the globe, habitat changes leading to a new ecology of zoonotic and phytopathogenic pathogens, and the impacts of antibiotic resistance and climate change, global health security will continue to be threatened by pathogen spillover and emerging disease outbreaks. For example, a recent CDC report shows that since 2013, cases of vector-borne human diseases in humans have increased by 300% [6]. 

Outbreaks such as that of the 2014–2016 Ebola virus in West Africa have led to the generation of potentially tens of thousands of samples by health care workers and epidemiologists, many of which remain unaccounted for [7]. Due to the public health emergency and the subsequent demand for sample material and data, the local capacity for ethical review boards and material transfer were overwhelmed, which resulted in the improper removal and tracking of samples out of West Africa by international partners. The safeguarding, proper handling, and disposal of the plethora of samples will continue to be challenging. The SFAF discussion panel recognized that in the current age of the internet and political influences, maintaining trust in sources of information is becoming more difficult. In the face of ever-changing geopolitical factors, trust and its resilience among partners ultimately influences the funding for international programs—especially those supporting capacity building at the local level. This local capacity for ensuring biosafety, biosecurity, and public education is too often quickly exceeded during an infectious disease outbreak. 

Building genomic sequencing capability through shared research projects and associated mentoring and training has been a focus of US international cooperative engagement. With the increased sequencing capacity and the greater availability and democratization of the technology, relevant metadata may replace physical samples due to its affordability and ease of use. Diminishing the risks associated with sample or pathogen exchange such as concerns for the loss of ownership and proprietary information can encourage transparency and knowledge sharing. Sequencing data can be shared without challenging political sensitivities often associated with exchanging physical materials such as samples or isolate material. However, in addition to initial instrument and reagent investments there is the additional challenge of storing and managing large datasets. Furthermore, there is still the potential for the unauthorized or nefarious access and misuse of genomic data. Largely due to the increased use of sequencing in biosurveillance and diagnostics, biosecurity has now expanded to include cybersecurity. Furthermore, the emergence of gene editing technologies such as clustered regularly-interspaced short palindromic repeats (CRISPR) and synthetic biology generates new challenges with respect to monitoring and defending against the use of technology for nefarious purposes. 

Significant progress has been made in the past five years to establish sequencing capabilities in partner country diagnostic and public health laboratories around the world. Both sequencing and bioinformatics technologies are becoming easier to use, faster, and most importantly, cheaper. Most cooperative biological engagement programs have the central mission of reducing the threat of pathogens of security concern through strengthening the capacity and capability to detect, diagnose, and report infectious disease outbreaks. Sequencing can help in each of these three areas, but in particular, it can play a critical role in disease diagnosis. Too often in all countries around the world the causative agents of an infection are never identified. Often, the lack of knowledge of whether an infection is due to bacteria or viruses can lead to the misuse of antibiotics, potentially facilitating more antibiotic resistance. The cause of a disease outbreak, not only in humans but particularly in animals, can remain unclear for weeks as diagnostic laboratories work through assays of possible pathogens. Next-generation sequencing can quickly discern all microbes in a sample, including viruses, if both RNA and DNA are used in extraction and sequencing. 

## 4. Conclusions: The Case for Continued Research and a Concerted Global Effort

Next-generation sequencing (NGS) technologies catalyzed a genomics revolution that collapsed the time required to characterize and identify the causative agents of disease outbreaks, as well as emerging and re-emerging infectious pathogens, from weeks to hours [8,9,10]. Digital communications and storage strategies have extended this revolution by enabling globally dispersed collaborators to share genomic information generated anywhere in the world, without the need to physically ship samples. 

Two current operational challenges for using sequencing in biosurveillance include standardization (of methodologies, bioinformatics, and reporting) and sustainability (of the technology in laboratories). Most of these international scientific assistance programs have identified ways to harmonize standards through increased cooperation and coordination among programs and country partners. However, assuring the long-term sustainability of laboratory capabilities may be the most difficult issue of all. Historically, sequencing technologies have required expensive annual service contracts, in-country technological expertise to run the sequencers and analyze data, and vigilance in maintaining the technology and bioinformatics platforms. Too-often, laboratories and partners are excited about bringing a new technology to a country or region but do not have a viable sustainability plan in place. Countries and laboratories must be able to support the additional costs and training required for new sequencing technologies. The additional costs, associated workforce training, and equipment maintenance must be discussed and agreed upon prior to establishing such capabilities in a laboratory. 

Infectious disease outbreaks continue to occur worldwide, with microbes moving to new regions and hosts and new transmission opportunities emerging. The evolution and ecology of infectious diseases form the nexus of several complex systems—the environment (including the role of climate change in altering the environment), humans (movement, sociology, politics, genetics, and cultural practices), livestock, wildlife, crops, vectors, and mitigations (vaccines, antibiotics, and trade). Understanding how these systems are changing and employing a multi-sectoral approach are critical for biosurveillance planning; knowing why outbreaks occur can lead to better detection and greater prevention potential. While research in support of biosurveillance activity may not have yielded defined and timely outputs, the return on investment has been proven repeatedly and may lead to important rewards. There is also a degree of “return on relationships” that results from reciprocating research activity involving international collaborations and networks. For example, Fair et al. highlighted numerous successes that followed a workshop that trained participants on the biosurveillance of bats [11]. The diverse and numerous collaborations from the initial workshop were mapped in a systems dynamics modeling framework that is often used in epidemiology, which was used to measure the relationship networks and outcomes. In a special Frontiers’ Topic on biological cooperative engagement, 24 articles with over 145 authors described how sequencing, genomics, and related research activities had been applied to enhance biosurveillance [12]. For example, Cui et al. [13] discussed the value of genomics in biosurveillance and a phased approach for establishing genomics centers in cooperation with partner countries. This phased approach provides an order of activities to build the capability and flexibility that are responsive to the evolution of technologies and business practices, reducing risks in achieving sustainability. In an example of research activities, Hay et al. [14] detailed the challenges, opportunities, and successes of conducting cooperative research on the biosurveillance of ticks and tick-borne diseases in Central Asia. 

While there are additional considerations when working with countries that may be deemed less collaborative, these countries may in fact be the ones where the most support is needed. It is also important to note the reasons why some of these countries do not exhibit more scientific transparency; often their concerns are related to the potential attribution or loss of proprietary material and their inability to account and control for that material. Working cooperatively with a shared mission to improve global health security not only builds technical capabilities within countries, but more importantly strengthens human relationships and thus trust between countries. By definition, cooperation builds trust and empathy for understanding our cultural differences and increases the transparency for sharing information and ideas. The 2014 Ebola outbreak in West Africa taught us about the importance of trust, empathy, vulnerability, and the challenges of working together. Countries that are less collaborative in nature may be more willing to cooperate with peers from other countries or through intergovernmental organizations, which is particularly important for the surveillance of infectious diseases and other biothreats. Policies should not limit the cooperation among countries when it comes to health security and newly emerging diseases or pathogens of security concern. The sharing of information, particularly pathogen genetic data from sequencing, will only become more important in the future, as demonstrated by recent discussions regarding the Nagoya Protocol and the World Health Organization’s Pandemic Influenza Preparedness Framework, and will be a primary tool to understand and guard against infectious disease outbreaks. 

The nature and breadth of biological threats is substantial, and international scientific collaboration should continue to strive to democratize knowledge through transparency that promotes global health security. Sequencing and bioinformatics technologies, particularly when used as collaborative tools, can be network multipliers that provide critical support to disease surveillance and reporting efforts necessary to identify and mitigate infectious disease outbreaks. These technologies can also forge the backbone of collaborative international networks that not only improve the understanding of the evolution, ecology, and emergence of infectious diseases, but also promote trust, transparency, and the communication necessary for individual nations and the international community to prepare for and respond to emerging biological threats. 

## Figures and Tables

**Figure 1 tropicalmed-04-00078-f001:**
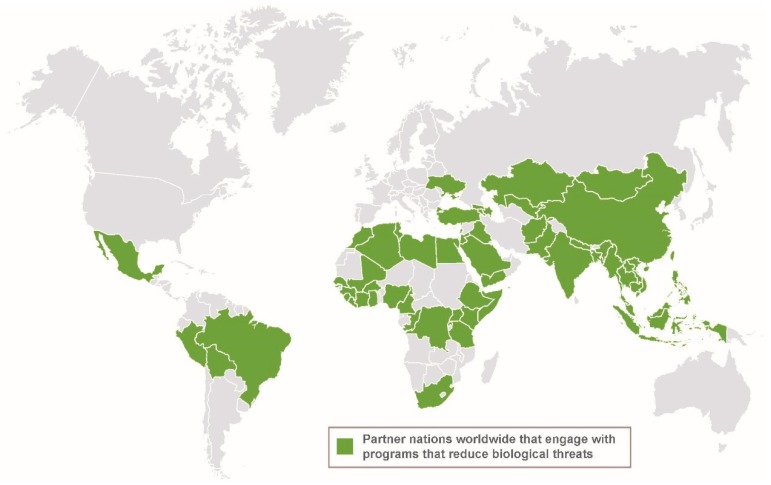
The governments of the US, Canada, Germany, and the United Kingdom fund programs that engage with over 40 countries worldwide to reduce biological threats. The countries represented are not necessarily engaged in current program activities.

**Figure 2 tropicalmed-04-00078-f002:**
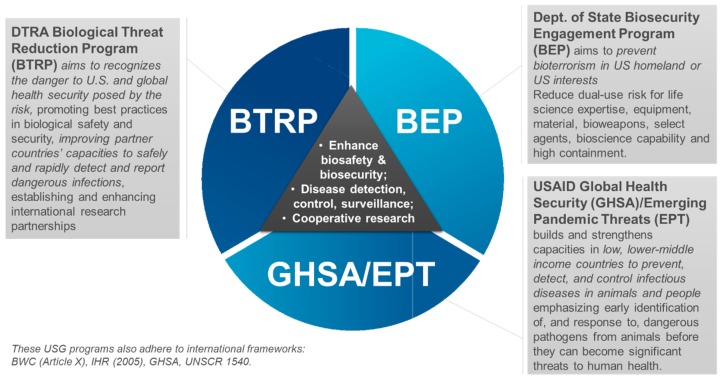
The aims and objectives of the United States (US) Department of State Biosecurity Engagement Program (BEP), the US Defense Threat Reduction Agency (DTRA) Biological Threat Reduction Program (BTRP), and the US Agency for International Development (USAID) Global Health Security Agency/Emerging Pandemic Threats (GHSA/EPT) programs.

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
