# Peer review of "Achieving Health Security and Threat Reduction through Sharing Sequence Data"

_tropicalmed, 2019, doi:10.3390/tropicalmed4020078_

Round 1
Reviewer 1 Report
Thank you for the opportunity to review this short communication.
It appears this group of scientists are trying to shoehorn the results of a panel discussion that took place during a recent SFAF conference into a short “Communication” for your journal. The goal is admirable and writing a coherent piece with so many authors is no easy task. In my opinion, rather than using the format of a scientific paper to espouse the (valued and valuable) views of the panel, a more discursive format would be in order, but I leave that to the publisher.
While the leanings of the authors are clear, I thought they did a reasonable job stating the risks and rewards inherent in supporting sequencing capability in non-transparent nations, even if that is often where the problems arise.
That said, the main point, that “Genomics-based international collaborations”
(a term I have never heard used before) can be a valuable asset in a crisis or outbreak situation, seems like a reasonable, if obvious, point and the authors make valid supporting arguments for their position.
Here are a few specific points:
The title is opaque and should be re-worded:
Blending Sequencing and Scientific Engagement:
Achieving Biological Threat Reduction and Global
Health Security
Given that, the “blending” of scientists and genomics is not exactly like “blending” a margarita; I would suggest choosing another title, or the authors risk confusing their readers.
Something similar, but more direct might help:
Using Shared Sequence Data to Help Achieve Health Security, or
Can Shared Sequence Data Help Achieve Health Security?
In the body of the communication, the authors mention the risks associated with locating technology and sharing data with non-transparent nations. The risks were given relatively little weight, and a more balanced treatment might give the publication more scientific gravitas, if the authors so choose.
Figure 2 was a nice diagrammatic summary of the tripartite roles of the various government efforts in this area.
The final line is weak and should be re-worded. Yes, the “democratization of knowledge IS exciting”, but this is not what the paper is about, and the phrase does not capture the rapid pace of change in this area, nor the ongoing benefits of international scientific collaboration.
Panel discussions often open up more questions than they answer, so my sympathies are with the authors, and recommend the publication of this report, with the small changes listed above.
Author Response
please see uploaded document.

Reviewer 2 Report
The approach is a little unusual in that the primary focus/purpose of the authors is to support 'international conferences and training events'...and the "democratization of knowledge is exciting." I would have thought there might be more emphasis on the value of the science discussed which has led to human relationships of trust and thus to threat reduction and global health security. If you look at the last sentence of the abstract and the last sentence of the paper the reader is led to believe that the importance is in international conferences, not in the good work done in collaboration with partners to improve sequencing capabilities...and outcomes of scientific engagement...as the title might suggest.
63---'redirecting' might be more accurate than 'eliminating' here
63 and 65----'securing biological WMD' does not apply to all the countries/regions listed in line 65
68---The overall mission of cooperative programs discussed here is narrower than 'to reduce the threat of infectious diseases'....if you read the mission statements of either BRRP or BEP.
78---I don't see that SFAF is defined before this use. Maybe I missed it.
105---Why '20' here and '25' on line 94?
106-113---This section could be written more clearly. It feels like the authors just wanted to list as many variables impacting disease emergence as possible and the result feels like 'apples and oranges' to some degree. ...and the statement that 'infectious diseases are evolving as a result' seems a little odd at the end. Is 'evolving' what you want here? And does this mean all 'infectious diseases'?
124-127---"trust" between whom? "stability"? Are the 'external factors' all the noice we here from Russia re the Lugar center. That whole section from about 120-127 could use some work. The Russian claims have been so outrageous as to make them almost meaningless. I'm not sure that this is the example to use in evaluating or supporting the importance of "trust".
128---"Building a genomic sequencing capability..."
129---"With increased sequencing capacity..."
134---"...as samples or isolate material."
141---Maybe "purposes" rather than "reasons"
142---"...in the last five years to establish sequencing .."
147---"...report infectious disease outbreaks"
149---"...the causative agents of an..."
150---Here I would probably use 'infection' rather than 'outbreak'. I may be wrong, but believe that misuse of antibiotics---to treat a viral infection or malaria for example---more commonly occurs at the individual patient level; not the 'outbreak' level.
159---'...and storage strategies have extended...'
166---Maybe use 'sustainability' in place of 'endurance' here.
183+---I don't think you do justice to Jeanne's good and (rare paper) here. RoR in this context is about the human relationships of trust, which doesn't really come out in the way it is presented. There's really no mention of that, so the average reader is likely to just skip over the sentence.
190---The term "starting momentum" is a little awkward. Maybe, '...this phased approach provides an orderly sequencing of activities to build capacity....'
194-~203---I think you need to be either careful not to offend current or future engagement partners or more clearly define what you mean by "security risk" countries and "sensitive" countries and 'countries of greater concern'. I think it's very important that you clarify this whole section.
197---"....strengthens human relationships...'
213-214---I was surprised by the last sentence and think it could be improved. It refocuses on 'the panel' and the processes again instead of the important issues discussed in the paper. It leads the reader to think this great group of authors is as interested in more panels as it is in TR and GHS. It's not too remarkable that the panel found the 'threats currently unprecedented'...although there wasn't a lot of time spent discussing threats in the paper. Finally, that the panel found "democratization of knowledge exciting" isn't that helpful either in dealing with TR and GHS. Therefore, I think the last sentence of both the abstract and the paper could be strengthened.
This is a very important paper because it presents a slice activities of the USG that are poorly understood and appreciated, not only on the Hill, but throughout the BD/ID community. I'm glad you guys have taken this on and think, with just a little more thought and work, it can be a very useful paper.
